# Comparative Transcriptomic Analysis Reveals Adaptive Traits in Antarctic Scallop *Adamussium colbecki*

**Samuele Greco** [1], **Giacomo Voltarel** [1], **Anastasia Serena Gaetano** [2], **Chiara Manfrin** [1], **Alberto Pallavicini** [1], **Piero Giulio Giulianini** [1] and **Marco Gerdol** [1,*]

1   Department of Life Sciences, University of Trieste, 34127 Trieste, Italy
2   Department of Chemical and Pharmaceutical Sciences, University of Trieste, 34127 Trieste, Italy
*   Correspondence: mgerdol@units.it

**Abstract:** Antarctica is the most extreme continent of Earth, with strong winds, freezing temperatures on land, and ocean temperatures constantly below 0 °C. Nonetheless, the Antarctic Ocean is home to an astounding diversity of living organisms that adapted to the multiple challenges posed by this environment via a diverse set of evolutionary traits. Although the recent advancements in sequencing technologies clarified the molecular bases of such adaptations in Antarctic fishes, little information is available for Antarctic invertebrates. In this preliminary study, we address this knowledge gap with a comparative transcriptomic approach to obtain insights into some of the adaptations that allow the Antarctic scallop *Adamussium colbecki* to survive and thrive in the freezing waters of the Antarctic Ocean. Despite some limitations, our analyses highlighted significant over-expression of genes involved in regulation of mRNA transcription, maturation, and degradation, which might compensate for a reduced efficiency of these processes at low temperatures. Other alterations detected in the Antarctic scallop transcriptome include enhanced expression of genes that regulate degradation of misfolded protein products and allow maintenance of cytoskeletal structure and function at subzero temperatures. Altogether, these observations support the presence of multiple previously unreported molecular adaptive traits in *A. colbecki*, which have important implications for our understanding of adaptation of this important component of the Antarctic trophic chain to such an extreme, but stable environment.

**Keywords:** antarctica; adaptation; comparative transcriptomics; mRNA splicing; cytoskeleton; protein misfolding; ubiquitination

**Key Contribution:** Through a comparative transcriptomics approach, this study provides for the first time evidence supporting the presence of molecular adaptations to cold in the Antarctic scallop *A. colbecki*, revealing significant over-expression of genes associated with different biological pathway that may counterbalance the detrimental effects of low temperatures on mRNA, protein, and cytoskeleton dynamics.



## 1. Introduction

Antarctica is the coldest and one of the windiest and most inhospitable regions on planet Earth, consisting of a land-mass continent covering about 14,000,000 km$^2$, surrounded by the Southern Ocean (SO). The SO represents one of the most unique marine environments of our planet. Water temperatures range from −1.86 °C (the freezing point of seawater) to +1 °C, with little variation across seasons, regardless of depth [1,2]. These conditions, which are thought to have been stable for more than 15 million years [3,4], are home to a vast biodiversity that includes many stenothermal organisms, adapted to this extreme, yet very stable, environment.

The endemic Antarctic scallop *Adamussium colbecki* (Smith, 1902) [5] is a bivalve species, which represents one of the key components of the Antarctic benthic community, and the

sole survivor of a once-diversified Pectinidae assemblage from the early Quaternary [6]. It is distributed in dense, genetically isolated clusters [7], with a fragmented distribution along Antarctic coasts. The lack of *Adamussium* populations in regions such as Victoria Land and McMurdo Sound can be explained by the presence of communities that feed on its larvae and by the thinness of its shell, which lowers the energy budget required for movement, but on the other hand exposes this species to physical damage from iceberg scouring [8]. Given its narrow temperature range [9] and sensitivity to physico-chemical changes in the environment, the Antarctic scallop has been proposed as a sentinel organism for monitoring the effects of changes in ocean waters [10].

*A. colbecki* can be found on soft and muddy bottoms, or on rocky substrates close to the abyssal portion of the shore [6]. It is typically found at depths of around 100 m [11], but it has a wide bathymetric range, as living individuals have been found beyond 1300 m below sea level [8]. Adult individuals are usually 70 to 75 mm long and are capable of movement by valve clapping, whereas juveniles are byssally attached to the shells of adults until the age of 3 to 5 years [12].

As with the other scallops, *A. colbecki* individuals can generate water jets, resuspending the debris in the surrounding area by clapping their valves [13,14]. Food particles are then trapped in a mucous substance present on the surface of the gills, moved to the mouth by cilia, and finally processed by the digestive gland [15].

As previously stated, *A. colbecki* has round, paper-thin valves, which present radial ribs providing strength to the shell [15]. As a consequence, the shell-to-wet body mass ratio is noticeably lower than other scallops living in temperate waters, decreasing the overall body density and suggesting that *A. colbecki* is significantly more buoyant in seawater compared with other scallop species of the same size [16].

The adaptive traits that allowed *A. colbecki* to be the only representative of the family Pectinidae living in the Antarctic Ocean are still poorly understood, but are believed to include low metabolic rates [17] and higher transcriptional activity compared with other scallops [18]. Moreover, the shell of *A. colbecki* displays a peculiar nanostructure that prevents the formation of ice sheets [19]. To date, while extensive scientific literature has been produced about the molecular strategies of cold adaptation used by other marine Antarctic metazoans, with particular reference to notothenioid fishes [20], little to no knowledge is available on this subject for the Antarctic scallop. This lack on knowledge is partly ascribable to the limited amount of -omic resources available, as the first and only reference transcriptome for this species was published in 2019 [21] and recently updated by our group [22]. Based on the availability of new gene expression data from this recent investigation, we used a comparative transcriptomic approach to obtain novel insights on the molecular adaptations that may have allowed *A. colbecki* to be the only surviving scallop species in the Antarctic Ocean.

## 2. Materials and Methods

### 2.1. Sampling, Sequencing, and Assembly of A. colbecki Transcriptome

Biological information about the individuals, methods for their sampling, sequencing, and assembly of the reference transcriptome, have been described in detail in our recent work about the transcriptomic response of *A. colbecki* to moderate thermal stress [22]. However, to maximize the sensitivity of our analysis (as described in detail below), the transcriptome assembly used as a reference in this work was an intermediate version, obtained prior to clustering and removal of redundancy.

### 2.2. Molecular Data from Other Pectinids

The reference genome assemblies from the non-Antarctic scallops *Argopecten irradians* [23], *Mizuhopecten yessoensis* [24], *Azumapecten farreri* [25] and *Pecten maximus* [26] were retrieved from the NCBI genome database. Matched RNA-seq reads from gills, digestive gland, and mantle [22,24,27–33] were downloaded from the SRA repository (see Table S1 for accession IDs and references, refer to original publications and/or related biosamples for further

biological information about these individuals). Unfortunately, we could not retrieve high-quality reads for gills of *P. maximus* and digestive gland of *M. yessoensis*. These four scallop species were selected due to the availability of high-quality genome sequence references, as well as due to their close phylogenetic relatedness with *A. colbecki*. Indeed, all the species belong to the family Pectinidae, whose latest common ancestor can be dated back to the Devonian period [34] and who are classified within three distinct subfamilies. Namely, *A. colbecki* is a member of Palliolinae, *M. yessoensis* and *A. farreri* are members of Pedinae, and *P. maximus* and *A. irradians* are members of Pectininae [21].

### 2.3. Identification of Orthologous Sequences

To ensure the reliability of the differential expression analysis, our comparative study was restricted to single-copy orthologous genes found in the genomes of all the non-Antarctic scallops of interest. This gene subset was identified through the reciprocal best hit (RBH) approach, by running pairwise similarity searches in the cDNA sequences extracted from genome annotations with blastn v.2.10.1 [35], setting the e-value threshold to $1 \times 10^{-10}$ and the percentage of identity threshold to 80%. This strategy was developed to take into account the genomic peculiarities of different species, as well as the presence of lineage-specific gene duplications and losses, which may hamper the interpretation of gene expression data. Although different approaches may be used to mitigate this issue, a set of genes or transcripts unequivocally identified as orthologs can be used as a reference for read mapping. This strategy was initially suggested by early comparative studies conducted on microarray data [36] and has been consistently used in recent times for the analysis of RNA-seq data [37–40].

One limitation of this setup lies in the fragmentation of some contigs included in the de novo assembled transcriptome of *A. colbecki* and in its high degree of redundancy, due to the large number of assembled, alternatively spliced isoforms per gene. To overcome such limitations, the orthologs from this species were retrieved using a refined method. Specifically, *A. colbecki* transcripts were used as queries for a diamond blastx v.0.9.14 [41] search (using the same thresholds reported above) against a database including the protein sequences of *M. yessoensis* and *P. maximus*. These species were selected due to the high completeness of associated gene annotations, compared with *A. farreri* and *A. irradians*, as evaluated by BUSCO [24,26]. All *A. colbecki* contigs sharing matching single-copy orthologous genes as top hits in the two genomes were added to the dataset.

Since the 3′ tag library preparation sequencing strategy used for *A. colbecki* had a strong 3′ bias [42], most reads were expected to lie in the 3′ UTR of the transcripts, close to the poly-adenylation site. Hence, in case of fragmentation, some *A. colbecki* contigs covering only the 3′ UTR region would be expected to be missed by the aforementioned blastx approach. To overcome this issue, we extended all the annotations of the set of orthologous genes in the four non-Antarctic scallops by 500 nt at the 3′ end. This strategy allowed us to recover the 3′ end of the transcripts from these species (or part of it), which is often missed by current gene annotation pipelines, mostly focused on the correct identification of the coding sequence. The transcriptome of *A. colbecki* was used as a query for a blastn run against this improved sequence dataset, enabling the retrieval of additional hits from this species. For the sake of clarity, the final sequence dataset included multiple transcripts derived from *A. colbecki* and a single reference transcript derived from each of the four other species. This sequence collection was used as a reference for all subsequent analyses.

### 2.4. Annotation of the Orthologous Sequence Dataset

The collection of orthologous transcripts was annotated using the pfam-clans branch of the annot.aM pipeline (https://gitlab.com/54mu/annotaM, accessed on 14 February 2023), which retrieved the Gene Ontology (GO) annotations from the best hits within Uniprot and Orthodb (MolluscaV10), and the Pfam annotations from the pfam database [43]. The annotations obtained from multiple *A. colbecki* isoforms were then merged and referred to the corresponding orthologous gene.

### 2.5. Quantification of Gene Expression

Another issue that we faced during the analysis lied in the difference between the methods (i.e., library preparation protocol, read length, etc.) by which RNA-seq data were generated in the different species. To assess the impact of these factors, we mapped all the sequencing reads against the reference with bowtie2 v2.5.1 [44] and extracted the mappings with samtools 1.16.1 [45], which allowed the production of the respective coverage graphs. The visual inspection of these graphs, paired with the calculation of scaled relative average mapping coverage for each species, allowed us to establish that a 400-nt region comprising the $3'$ end of each transcript was the most appropriate window of analysis to ensure full compatibility among species, attenuating the technical biases linked with the use of different library preparation protocols. Therefore, gene expression levels were calculated on the number of reads mapped to the last 400 nt of each transcript only, disregarding the remaining part of the reference sequences. Read count tables were therefore generated for the aforementioned regions of the reference sequences and used as an input for the following differential gene expression (DGE) analysis. Despite the use of different library preparation protocols for *A. colbecki* and other scallop species (i.e., $3'$ tag vs. standard Illumina libraries), based on literature data, this technical factor was not expected to lead to significant systematic biases [46,47].

### 2.6. Differential Gene Expression Analysis

While the read count table from *A. farreri*, *A. irradians*, *M. yessoensis*, and *P. maximus* did already include a single representative transcript isoform per gene, the read counts obtained from the multiple transcript isoforms of *A. colbecki* were merged and redirected to the putative orthologous gene inferred as described above. Read count data were then loaded in an R v.4.2.2 [48] environment, alongside a metadata table including the species names and tissue of origin for each sample. Subsequent analyses were performed separately for each of the three tissues (i.e., gills, digestive gland, and mantle). A first edgeR v.3.38.4 [49,50] run was performed to extract the residuals from the fitted model on upper quartile-normalized [51] counts, which were used to remove background noise with RUVSeq v.1.28.0 [52], using the species as a grouping factor (RUVr function). A second run of edgeR was performed on the denoised, TMM-normalized [53] data, extracting all significantly up-regulated genes (URGs) based on thresholds of False Discovery Rate [54] (FDR) < 0.05 and logarithm of fold change (logFC) > 0. Those detected in all pairwise comparisons between *A. colbecki* and the four other species were kept for further analysis. All the down-regulated genes were disregarded, as it was not possible to determine whether the low expression observed in the Antarctic scallop represented a real biological signal or was due to the incomplete assembly (or missed detection) of the $3'$ UTR region.

### 2.7. GO Enrichment Analysis of URGs

For each tissue, the background set of genes used for DGE analyses was selected based on a minimum count of one mapped read in at least one species, in order to retain only the genes displaying biologically meaningful expression in each tissue and to avoid biases in the enrichment analysis [55]. Due to the overlap of the results obtained in the three tissues (see the Results section), the enrichment test was also performed on the subset of URGs shared by the three tissues, using the genes expressed in all tissues as the background. The analysis was performed with a hypergeometric test according to the methodology defined by Falcon and Gentleman [56] and terms were considered as significantly enriched when they displayed a FDR-corrected *p*-value < 0.05 and an Observed-Expected value $\geq 2$.

### 2.8. Further Analysis of URGs

The URGs shared among the three tissues were carefully reviewed manually, by recovering the associated gene function from Uniprot [57]. For the few genes lacking a BLAST hit against Uniprot, the protein sequences were submitted to the Interoproscan [58] online service, inferring their most likely function through the identification of conserved

domains. This set of shared DEGs was also analyzed on string-db [59], in an attempt to build an interaction network and obtain insights on co-regulated and interacting genes that were not included into the set of orthologous genes. To this end, the annotations of our dataset were projected to those of their human orthologs (in case of multiple hits in the annotation, all hits were used), and uploaded to the string db website. A network was then built and evaluated using all the expressed URGs as the background. The top 10 interactors were then added to each node (whenever available) with a confidence threshold of 0.4. Finally, the analysis function of string DB was exploited in order to obtain insights on the functional enrichment of the network, based on the over-representation of terms from Gene Ontology, KEGG, Reactome, and UniProt keywords.

## 3. Results

### 3.1. Identification and Annotation of Orthologous Sequences

The RBH method allowed the identification of 2346 orthologous genes shared by the four non-Antarctic species, then at the same time also unequivocally matching at least one transcript from *A. colbecki*. Within this set of sequences, 2235 (95%) displayed a Uniprot hit, and were therefore assigned GO terms (see Table S2 for detailed annotation).

### 3.2. Read Mapping

A plot of the average mapping coverage distribution along the final 400 nucleotides placed at the 3' end of each transcript, i.e., the region used to calculate gene expression levels, is displayed for each species in Figure 1.

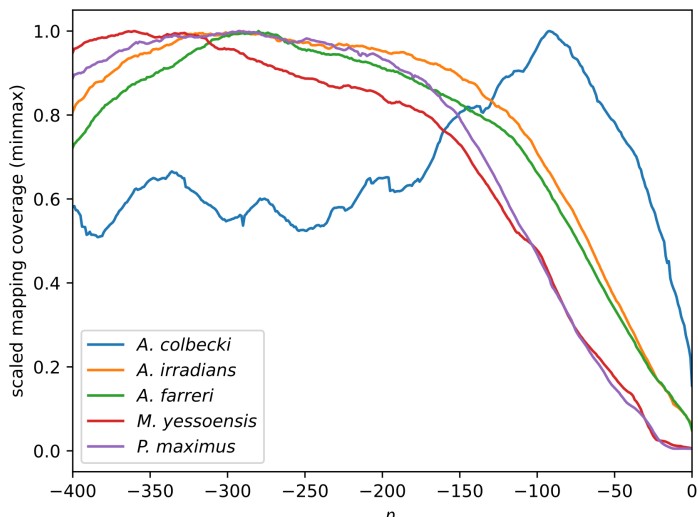

**Figure 1.** Scaled relative average mapping coverage of the RNAseq reads along the final 400 nt of the mRNAs obtained from the reference genomes of the four non-Antarctic scallops and from the transcriptome of *A. colbecki*. The *x* axis reports the position relative to the 3' end of sequences, whereas the *y* axis reports the minmax (minimum = 0, maximum = 1) scaled mapping coverage. The plot shows only the portion of mapping that was used in the downstream analysis.

This strategy was developed to allow for a reliable comparison between the data obtained from *A. colbecki* (with reads generated with a 3' tag library preparation protocol) and the other four scallop species (with reads generated with a standard poly(A) selection library preparation protocol). Indeed, while the reads obtained from *A. colbecki* displayed, as previously reported by other studies [46], a higher mapping bias towards the 3' end of each transcript compared with those obtained from the four other species using standard library preparation protocols, this was compensated by a stronger drop in coverage between position −150 and −400. Through preliminary tests, we determined that a 400 nt window

of analysis was associated with the lowest mapping bias compared with slightly shorter or longer-length intervals.

Following this procedure, *A. colbecki* displayed a significantly lower average number of mapped reads per sample, consistently with the lower sequencing depth required for 3′ tag approaches, compared with standard RNA-seq protocols [60] (Figure 2). The average mapping rates and standard deviations for all species are reported in Table S3.

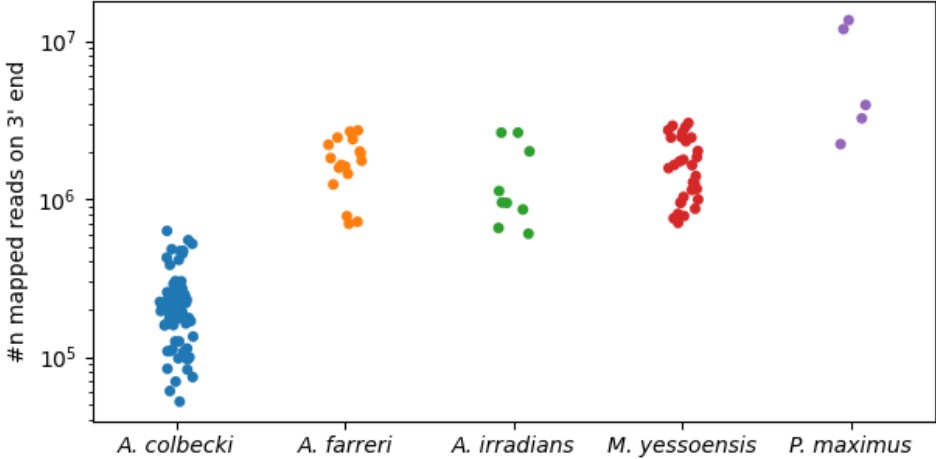

**Figure 2.** Per-sample number of reads mapped on the final (3′) 400 nt of the reference sequences, divided by species and log-scaled.

*3.3. Differential Expression and GO Enrichment Analysis*

The first run of edgeR allowed the identification of residuals used for the normalization and the detection of eight *A. colbecki* samples with anomalous expression profiles (the very same samples had been also previously identified as outliers in their original study [22]). As we had previously considered such samples as the product of high RNA degradation occurring during the transportation of material from Antarctica to Italy, these were marked as outliers and disregarded from further analysis. Normalized read count data highlighted a very significant clustering of scallop RNA-seq datasets based on the species of origin for all the three tissues, as depicted by the MDS plot shown in Figure 3. Strikingly, the first MDS dimension nearly entirely explained the differences between *A. colbecki* and non-Antarctic scallops, whereas the second dimension explained most of the differences among the *A. irradians*, *A. farreri*, *M. yessoensis*, and *P. maximus*.

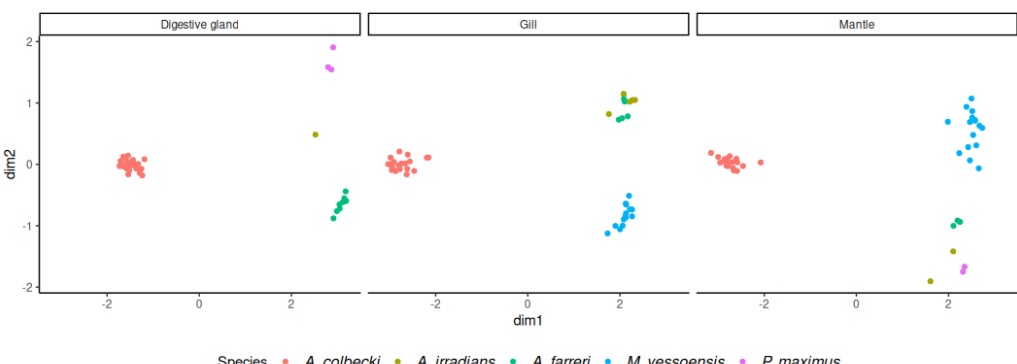

**Figure 3.** Scatter plots derived from the MDS analysis on the expression data derived from the three tissues, colored by genus.

A total of 237 URGs (i.e., genes significantly up-regulated an *A. colbecki* in all pairwise comparisons with the other scallop species) were detected in the digestive gland tissue, 245 in the gills, and 199 in the mantle. Of these, 127 were shared by the three tissues (Figure 4).

The results of all the pairwise DGE analyses between *A. colbecki* and the other pectinid species in the three tissues are included in Table S4.

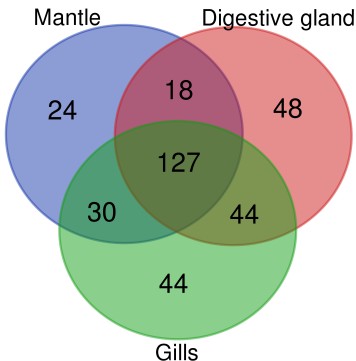

**Figure 4.** Venn diagram representing the sets of URGs found in the three tissues.

Enrichment analyses were conducted independently for each of the three tissues. A summary of the enriched Gene Ontology (GO) terms is presented in Table S5. As expected from the relevant overlap of the URGs sets, the examination of the enrichment results revealed a significant overlap in enriched GO terms among the three tissue types, indicating that the observed transcriptomic profiles could represent a general adaptation at the organismal level, largely independent from the specific tissue type. The overall consensus from the enrichment tests in the three tissues showed an activation of genes involved in transcriptional control, mRNA maturation and turnover, protein synthesis and degradation via ubiquitination (Figure 5). The full annotation of all URGs, together with simplified tags for the functions of each gene, is available in Table S7.

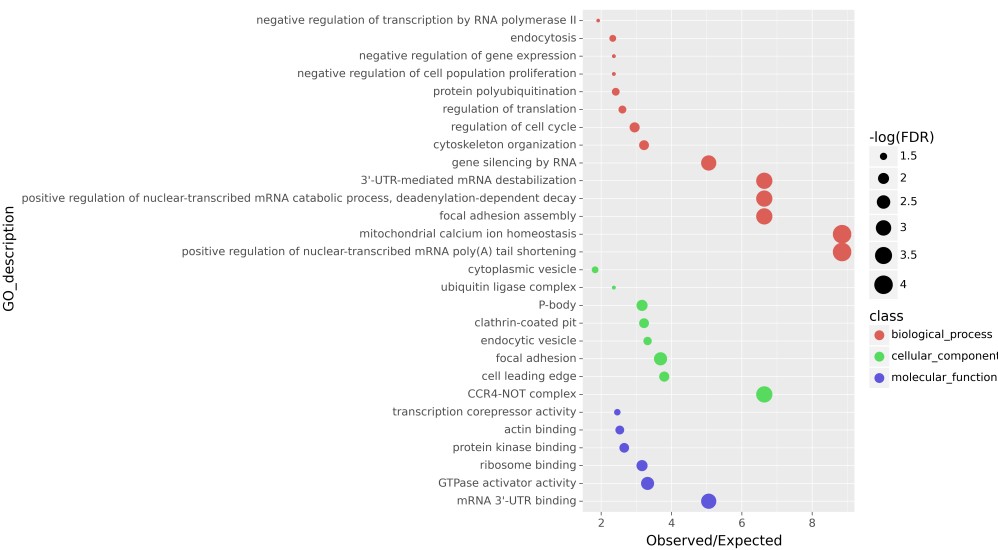

**Figure 5.** Enriched GO terms associated with the URGs shared by the three tissues. The size of the points represents statistical significance by $-log_{10}(FDR)$. Values equal to zero were set to $10^{-4}$ for visualization purposes. The position of the points on the $x$ axis represents their enrichment values and the color represents the Gene Ontology class.

The STRING network built with the URGs shared by all tissues had 135 nodes and 143 edges, with an average node degree of 2.12 and a mean clustering coefficient of 0.369. The expected number of edges was 108, and the PPI enrichment $p$-value was $7.05 \times 10^{-4}$, suggesting the involvement of biologically significant interactions. The string DB analysis resulted in the construction of the network shown in File S1. The results of the enrichment analysis of this network were in line with those of the GO enrichment analysis of the URGs,

reinforcing the idea that genes involved in mechanisms such as transcription regulation, mRNA maturation and turnover, and protein degradation were generally up-regulated in the Antarctic scallop, compared with other non-Antarctic pectinids (see Table S6 for detailed results).

## 4. Discussion

### 4.1. Limitations of the Study

As mentioned in the Methods section, the main limitation of this study lies in the different methodologies used for the generation of the RNAseq libraries of our species of interest. Indeed, the most evident consequence of such differences is the discrepancy between the mapping coverage distribution of *A. colbecki*, expected to mostly involve the 3′ end, and the other species, with a more uniform mapping along the full length of the mapping [46,47]. We adopted several pre-processing methodologies and strict selection criteria in order to mitigate this kind of bias, even though previous studies have demonstrated the comparability of gene expression datasets obtained with the two distinct library preparation protocols [46,47]. These include selection of the 400 terminal nucleotides for counts, normalization steps, and selection of only up-regulated genes, thereby pinpointing possible signatures of cold adaptation. Specifically, we expected the over-expression of such genes to be beneficial for improved survival in the freezing waters of the Antarctic Ocean. Nevertheless, while most of the identified up-regulated DEGs are likely to represent a real biological signature, we cannot exclude the possibility that some of these were outliers linked with technical factors, such as the superior ability of detection of short transcripts by 3′ tag libraries compared with standard libraries [46].

Although we have previously shown that the reference transcriptome, obtained from multiple adult tissues, displayed a high level of completeness [22], several transcripts subjected to strict regulation were likely not represented, a result consistent with previous transcriptomic investigations conducted on other bivalves [61], preventing the investigation of their up- or down-regulation. Taking into account this limitation, our analysis allowed the detection of alterations of the expression of genes constitutively expressed in all *A. colbecki* tested tissues, but it might have missed other equally important molecular adaptations involving inducible genes, or those exclusively expressed during embryogenesis. Moreover, molecular adaptations to cold may also involve gene gain, gene loss, and gene duplication processes (like those observed in Antarctic notothenioid fishes [62]), which we could not investigate due to the focus placed on the identification of single-copy orthologs following the strategy that is most commonly used to carry out comparative transcriptomic studies [36–40,63]. Such a fine-scale analysis will be only possible when a high-quality annotated genome assembly will be released for this species.

### 4.2. Multiple A. colbecki Tissues Display a Shared Transcriptomic Adaptation to Cold

The increasing availability of molecular data has allowed the discovery of several adaptive traits in species living in the Antarctic Ocean, with a particular focus on notothenioid fishes such as *Chionodraco hamatus* or *Trematomus bernacchii* [37,62,64,65]. These molecular approaches provided information complementary to previously collected morphological, biochemical, and physiological data, increasing our understanding of the mechanisms underlying cold tolerance in Antarctic organisms. For example, such adaptations in notothenioids include the expression of antifreeze glycoproteins, encoded by an expanded family of trypsinogen-derived genes, which prevent ice-crystal formation in body fluids [66,67], or the loss of an inducible heat shock response, which facilitates protein folding at sub-zero temperatures [68]. Nonetheless, these scientific developments have not progressed at a similar pace in invertebrates, even though molecular approaches have already shown great potential in uncovering novel cold adaptation strategies, as in the case of Antarctic amphipods [69]. Although *A. colbecki* is the most abundant bivalve species in Antarctic coastal waters and a key component of the Southern Ocean food web, our current knowledge of the mechanisms that enable this species to survive and thrive at sub-zero

temperatures is limited to a handful of studies, none of which has exploited large-scale -omic approaches to date.

The comparative transcriptomic analysis that we carried out between the Antarctic scallop and four pectinid species living in temperate waters identified a core set of 127 URGs. These genes, which displayed the most consistent and solid evidence of over-expression across different tissues, are most likely involved in functions associated with processes of broad biological significance (i.e., not restricted to specific tissues), whose expression needs to be finely tuned in all body parts to maintain homeostasis in this stenothermal organism at sub-zero temperatures.

The remarkable fraction (~5.4%) of the orthologous genes shared by the five investigated pectinid species that were up-regulated in *A. colbecki* supports the notion that the transcriptional profiles of Antarctic and non-Antarctic scallops display remarkable differences, which are clearly highlighted by MDS analyses (Figure 3). Although we cannot rule out the existence of additional tissue-specific molecular adaptations to cold, the relatively small number of tissue-specific URGs identified by our comparative analysis (Figure 4) prompted us to investigate with greater depth only the functions associated with the set of URGs shared by all tissues.

### 4.3. The Antarctic Scallop Displays Improved Control of mRNA Transcription, Processing, and Turnover

Temperature is well known to affect RNA dynamics on different levels, even though such processes are regulated by complex relationships, which largely vary among species [70]. As with all other molecules endowed with catalytic activities, the function of important molecular players in the context of transcription, such as the RNA polymerase II and spliceosome complexes, is strongly influenced by temperature. Although many Antarctic enzymes show remarkable adaptations that enable improved functionality at low temperatures [71–73], it is currently unknown whether similar modifications also affect the molecular components of the machinery responsible for mRNA transcription in *A. colbecki*.

A high number of enriched GO terms associated with the URGs shared by all *A. colbecki* tissues (see Supplementary Table S1a) were involved in the maturation and degradation of mRNAs. In detail, annotations such as "positive regulation of nuclear-trainscribed mRNA poly(A) tail shortening", "3′-UTR-mediated mRNA deastabilization", "P-body", and "positive regulation of nuclear-trainscribed mRNA catabolic process, deadenylation-dependent decay" clearly support enhanced activity of the molecular machinery responsible for shortening the poly(A) tail of mRNAs, thereby leading to their subsequent degradation by exonucleases [74]. The key up-regulated orthologous genes involved in this process were CNOT3 and CNOT7, part of the CCR4-NOT complex, which is the major deadenylation factor in eukaryotes [75,76]. However, other important factors that may aid this process were also over-expressed in *A. colbecki*. Namely, these included PABPC1, which mediates the degradation of mRNAs containing premature stop codons [77], RNA helicase HELZ, which is a direct interactor with the CCR4-NOT complex [78], TARDBP, a multifunctional 3′ UTR binding protein that promotes mRNA instability [79], and GIGYF1, a key component of the 4EHP-GYF2 complex, which drives the degradation of target AU-rich mRNAs [80].

Moreover, another critical URG was the TNRC6A/GW182 homolog, which recruits deadenylase complexes to the poly(A) tail of mRNAs targeted by specific miRNAs [81]. This observation would also support the significant enrichment of the GO term "gene silencing by RNA", even though the existence of cold-inducible miRNAs, previously reported in other eukaryotes [82–85], remains to be investigated in the Antarctic scallop.

We interpret the presence of such remarkable adaptations as the possible result of the lower splicing efficiency attained at low temperatures in the Antarctic environment [86]. This may result in an increased rate of intron retention and, consequently, in the accumulation of aberrant transcripts. In addition to the requirements for the constant activation of mRNA decay pathways, this may also imply that *A. colbecki* needs to invest more resources, compared with its relatives living in temperate waters, in maintaining a highly

effective splicing machinery. This hypothesis would be consistent with the observed up-regulation of INTS2, a component of the integrator complex, responsible for the maturation of snRNAs [87]; G3BP2, which can stabilize or degrade mRNAs upon their binding [88]; SRRM1, an important splicing regulator [89]; and TCERG1, which coordinates transcriptional elongation with splicing [90]. Moreover, in addition to its role in mRNA decay, the aforementioned URG TARDBP is also involved in the recognition of UG-rich elements located in the long introns of pre-mRNAs, promoting their efficient splicing [91]. Finally, the protein Pasilla homolog, also found in the list of URGs in this study, may also play a role in the processing of pre-mRNA during the splicing process [92].

Nevertheless, an alternative interpretation of the alterations reported in the previous paragraphs is also possible. Indeed, the up-regulation of mRNA splicing and degradation observed in *A. colbecki* may be simply linked with a high abundance of mRNAs, rather than being indicative of a high rate of aberrant transcripts. Indeed, other authors have previously reported that the Antarctic scallop displays significantly higher mRNA levels and higher RNA/protein ratios than the temperate scallop *Aequipecten opercularis*, which may allow this species to rapidly up-regulate protein synthesis during peaks in primary production [93], counterbalancing the lower efficiency of translation expected at low temperatures [94]. The significant up-regulation of GPN1, a major RNA polymerase II nuclear import factor, could support this alternative hypothesis [95].

More efficient control of the mRNA transcription process also could be obtained thanks to the regulation of chromatin accessibility. Although the URGs associated with enriched annotations, such as "negative regulation of trancription by RNA polymerase II", "transcription corepressor activity" and "negative regulation of gene expression", could not point towards the regulation of any specific biological pathway, several DEGs encoded histone-modifying enzymes and other chromatin-remodeling factors. These included the histone methyltransferase ASH1L and KMT2C, plus the histone demethylase KDM7A, which all promote transcription [96,97]. The histone acetylase EPC1 (which can act both as a positive or a negative regulator of transcription) [98] and the deacetylase recruiting factor ANKRD12 were also significantly up-regulated. Moreover, a number of other chromatin-remodeling factors, whose specific role in bivalves cannot be ascertained due to their functional characterization in vertebrates, such as RSF1, BRD2, CBX3, INO80D and BCL11A, were part of the set of URGs shared by all *A. colbecki* tissues.

Despite the considerable number of molecular studies carried out in notothenioid fishes, processes related to mRNA synthesis and turnover do not frequently appear among those that have been implicated in cold adaptation. For instance, while early pioneering transcriptomic studies reported an over-representation of GO terms linked with the regulation of transcription in *Pagothenia borchgrevinki*, compared with *Danio rerio* [99], we failed to detect similar molecular signatures in a larger-scale comparative transcriptomic analysis [37]. Therefore, it appears that the up-regulation of genes regulating mRNA splicing and degradation may represent a unique feature of the Antarctic scallop.

### 4.4. Evidence of an Enhanced Control of Protein Production and Turnover

Protein folding is one of the processes most critically affected by extreme temperatures, which can therefore impair protein homeostasis, having a detrimental impact on cell function and structure [100]. Indeed, nascent polypeptidic chains undergo a particularly slow folding process at sub-zero temperatures, due to the existence of kinetic constraints on protein stability [101]. In addition, even correctly folded proteins may undergo a counter-intuitive denaturation process at freezing temperatures due to an enhancement of hydrophobic group solvation [102,103]. Altogether, these two factors are thought to lead to a higher rate of misfolded proteins and/or protein aggregates in Antarctic organisms.

The molecular adaptations to cold which concern the protein folding machinery have been extensively studied in Antarctic notothenioids, revealing that these animals, unlike their relatives living in temperate waters, have lost an inducible heat shock response. They display, on the other hand, a high constitutive expression of a broad range of heat

shock proteins and other molecular chaperones [104,105], which sometimes belong to expanded gene families [106]. Similarly, Antarctic fishes display significantly higher levels of ubiquitinated proteins and higher expression of genes encoding proteasome components compared with their non-Antarctic relatives [37,107]. These alterations most likely represent adaptations that counterbalance the aforementioned increased rate of protein misfolding, ensuring efficient protein turnover.

While it is currently unknown whether similar adaptation strategies are present in the Antarctic organisms, our comparative analysis identified "protein ubiquitination" and "ubiquitin ligase complex" as two of the most significantly enriched GO terms associated with the set of shared URGs. This finding would suggest that, in a fashion similar to notothenioids, *A. colbecki* displays a very high rate of misfolded proteins that need to be targeted to proteasomal degradation to maintain protein homeostasis [108].

Most notably, we evidenced the over-expression of a number of unrelated orthologous genes encoding ubiquitin-conjugating enzymes, which include UBE2R2, RCHY1, MAEA, UBE2H, BTRC, and FBXL14 (the latter two are components of the SKP1-cullin-F-box complex). Two other URGs, i.e., ECPAS, a proteasome-associated protein, and AMFR, an E3 ubiquitin-protein ligase part of the VCP/p97-AMFR/gp78 complex, are implicated in the endoplasmic reticulum-associated protein degradation (ERAD) pathway [109,110]. Finally, USP48, a protease that specifically targets polyubiquitinated proteins, helping to maintain steady-state protein levels, was also included in the set of URGs in the Antarctic scallop [111].

Altogether, these observations suggest a generalized increase in the targeting of misfolded proteins to the proteasome via ubiquitination in *A. colbecki*. However, unlike the case of Antarctic fishes, we could not identify a generalized increase in expression of molecular chaperones, with the exception of DNAJC25, a Hsp40 family member [112], which suggests that improved protein folding efficiency is not part of the molecular adaptations of the Antarctic scallop.

On the other hand, we identified the up-regulation of a few genes whose protein products play an important role in the sorting of nascent proteins to the correct subcellular compartment. In detail, the list of these URGs included SRP72, a component of the signal recognition particle (SRP) complex, and, most importantly, NACA, which associates with nascent polypeptides, preventing the mistranslocation of non-secretory proteins to the endoplasmic reticulum [113]. The alteration of expression of genes involved in the targeting of newly synthesized proteins has not been reported before in notothenioid fishes and may therefore be considered as an adaptive trait restricted to *A. colbecki*.

Gene expression data further suggest that a more efficient protein translation process could be obtained in *A. colbecki* through the over-expression of a number of ribosome-associated genes, as indicated by the significant enrichment of the GO terms "regulation of translation" and "ribosome binding". These observations are supported by the up-regulation of some genes involved in ribosome biogenesis, such as FTSJ3 (a RNA methyltransferase), DDX52 (a DEAD box RNA helicase), PA2G4 (a RNA-binding protein associated with pre-rRNA complexes), and WDR36 (a repetitive nucleolar protein functionally homologous to yeast Utp21) [114]. Moreover, improved transcriptional control could also depend on the up-regulation of the translation initiation factor EIF4E, as well as of its repressor EIF4EBP2 and of CPEB2, which is part of a family of factors that binds specific elements found in the 3′ UTR of mRNAs, either positively or negatively regulating their translation at the initiation and elongation phases [115]. Nevertheless, note the fact that previous studies have revealed that the protein translation machinery of *A. colbecki* has an energetic efficiency comparable with that of other eurythermal species [93], implying that the translation process itself is not cold-adapted.

In addition, we might hypothesize that, besides being merely linked with the lower efficiency of the protein folding process at low temperatures, the requirements for higher control of ribosomal protein synthesis and enhanced protein ubiquitination activity may derive from the high mRNA concentration found in *A. colbecki*. As previously mentioned, in

line with the interpretation provided by Norkko and colleagues [18], this might allow rapid up-regulation of protein synthesis in summer, when primary production in the Antarctic Ocean is higher, but on the other hand would lead to a great abundance of truncated protein products, linked with the translation of badly spliced mRNAs.

*4.5. Optimization of Cytoskeletal Function at Low Temperatures*

Cytoskeletal dynamics are expected to be strongly affected by low temperatures, since the microtubules of organisms living in temperate environments start to disassemble below 4 °C, preventing further elongation and thereby impairing the maintenance of cell structure, motility, and cell cycle progression [116]. Previous studies, conducted on Antarctic algae, ciliates, and notothenioids, have demonstrated that these organisms have cold-adapted tubulins, which display a number of convergent non-synonymous substitutions. These proteins are characterized by intrinsically slow polymerization dynamics [117] and have a high proportion of hydrophobic contacts, which promote efficient microtubule assembly at sub-zero temperatures [118–123]. Moreover, the duplication of *α*- and *β*-tubulin genes evidenced in notothenioid fishes is also suspected to be a major contributor to cold adaptation in these organisms [124]. While the presence of similar molecular adaptations has not yet been described in the Antarctic scallop, the highly conserved nature of the microtubule polymerization process implies that significant cytoskeletal modifications must be present in all cold-adapted organisms, including protostomes, as confirmed by the identification of cold-adapted tubulins in the annelid *Mesenchytraeus solifugus* [125].

In line with this hypothesis, we detected the presence of several enriched terms linked with cytoskeletal function associated with the set of URGs shared by the three tissues. Although tubulin genes themselves were not up-regulated, several other molecular components associated with the cytoskeleton (e.g., the GO tem "cytoskeleton organization") or connecting the cytoskeleton with the extracellular matrix (e.g., the GO terms "focal adhesion" and "focal adhesion assembly"), were significantly over-expressed compared with non-Antarctic scallops. One of these URGs, encoding the katanin-interacting protein KATNIP, is involved in maintaining microtubule function, promoting their repair in case of damage and regulating endosomal trafficking [126]. The up-regulation of MAP1B, fundamental for the correct assembly of microtubules [127]; C2CD3, which is required for centriole elongation [128]; and HYDIN, a cilia-associated protein [129], further support the presence of cold adaptations linked with microtubule assembly and organization in *A. colbecki*. Although we did not detect any evidence supporting the up-regualtion of tubulins, the future availability of a fully sequenced genome for *A. colbecki* might enable the investigation of whether these gene families underwent an expansion, as in the case of notothenioid fishes [124].

Moreover, the up-regulation of CAPZA1, a key regulator of actin filament growth, DLC1, an important regulator of actin cytoskeleton structure [130]; and SYNE1, which connects the actin cytoskeleton with the nuclear lamina [131], suggest that cytoskeletal alterations in the Antarctic scallop are not restricted to microtubules, but also involve actin filaments. The over-expression of FERMT2, a scaffolding protein that connects extracellular matrix adhesion sites and the actin cytoskeleton, thereby being a major component of focal adhesions, would further support this interpretation. Interestingly, dystonin, which orchestrates the organization and interconnection among intermediate filaments, actin filaments, and microtubules, was also part of the set of shared URGs [132].

Not surprisingly, due to the intimate link between cytoskeletal functions and cell division, a number of URGs have been previously described as key players in mitosis. Namely, RCC2 regulates kinetochore–microtubule interactions in the early phases of cell division [133]; STARD13 regulates cell replication by altering actin dynamics [134]; LLGL1 is involved in the determination of mitotic spindle orientation [135]; PDS5 is part of the cohesin complex, which regulates sister chromatids cohesion during meiosis [136]; and FZR1 is a critical regulator of APC/C activity during anaphase and telophase [137]. RAC1, one of the main interactors of RCC2, was also one important URG, as it showed a large

number of contacts in the STRING network (File S1). This gene codes for a small GTP-binding protein that has an important role in regulating several cellular functions, including the maintenance of microtubule stability and the reorganization of actin filaments. Being a signal transducer, this gene also has implications in the activation of transcription and control of chromatin [138], two processes that have been extensively discussed above. A similarly central role can also be extended to the URG MAPK3, another signal transducer mainly involved in the regulation of cell cycle and cytoskeletal remodelling [139]. Cyclin D2, which regulates the G1/S transition during mitosis, was also included in the set of URGs.

Other up-regulated orthologous genes were linked with vesicular transport, as highlighted by the significant enrichment of several GO terms ("cytoplasmic vesicle", "endocytosis", "endocytic vesicle" "clathrin-coated pit"), often connected with the motor activity of GTPases (associated with the GO term "GTPase activator activity"). These included AP2A1, CLSTN1, FNBP1, GAPVD1, GLC1, HATR5B, RAB35, RAB43, SAR1B, SEC24B, and USP6NL. Although it is presently unclear whether the generalized up-regulation of endocytosis-linked genes in *A. colbecki* is linked with altered cytoskeletal dynamics, it is important to note that we have previously made a similar observation in notothenioid fishes, which also display constitutively higher levels of expression of genes involved in clathrin-mediated endocytosis compared with non-Antarctic species [37].

## 5. Conclusions

In conclusion, the comparative transcriptomic analysis between the Antarctic scallop *A. colbecki* and its temperate counterparts *Pecten maximus*, *Mizuhopecten yessoensis*, *Argopecten irradians*, and *Azumapecten farreri*, provided the first insights into the molecular traits of cold adaptation acquired by this species to survive in the freezing waters of the Antarctic Ocean. The three tissues analyzed in this study displayed a shared set of up-regulated genes, which therefore emerge as candidates for explaining the remarkable cold tolerance of the Antarctic scallop. Namely, the identified URGs were mostly involved in the maintenance of mRNA and protein homeostasis, as well as in the preservation of cytoskeletal functions. Although some of these adaptations are shared with other Antarctic marine metazoans (e.g., a higher level of protein ubiquitination and proteasomal degradation, most likely linked with increased protein misfolding rates), others, such as an enhanced efficiency of the mRNA splicing and deadenylation processes, have no known parallel in other Antarctic species. Moreover, we highlighted the presence of mechanisms, based on increased constitutive expression, that might aid to maintain cytoskeletal dynamics at subzero temperatures in a manner complementary to the previously described sequence mutations of Antarctic tubulins. Altogether, the results of this study offer new insights into the molecular basis of cold adaptation in marine organisms and have important implications for our understanding of the mechanisms of environmental stress tolerance in polar regions.

**Supplementary Materials:** The following supporting information can be downloaded at: https://www.mdpi.com/article/10.3390/fishes8060276/s1, Table S1: sources and references for the data used in this work; Table S2: Complete annotations of the orthologous genes set for the five species of interest; Table S3: Mean mapping rate and standard deviation for the five species; Table S4: Results of all the differential expression analyses in all tissues; Table S5: Results of GO enrichment tests for (a) overlapping URGs from all tissues, (b) URGs from gills, (c) URGs from digestive gland, (d) URGs from mantle; Table S6: Results of the STRING network analysis; Table S7: Annotation of the overlapping URGs; File S1: STRING network in cytoscape format.

**Author Contributions:** Conceptualization, S.G. and M.G.; methodology, S.G., G.V., A.S.G., C.M. and M.G.; formal analysis, S.G., G.V. and M.G.; resources, P.G.G., M.G. and A.P.; data curation, S.G., G.V. and M.G.; writing—original draft preparation, S.G., G.V. and M.G.; writing—review and editing, S.G., G.V., A.S.G., C.M., A.P., P.G.G. and M.G.; visualization, S.G.; funding acquisition, P.G.G., A.P. and M.G. All authors have read and agreed to the published version of the manuscript.

**Funding:** This work was funded by the Italian Program of Antarctic Research (Grant Nos. PNRA16_00099, PNRA16_00234 and PNRA18_00077).

**Institutional Review Board Statement:** The sample collection and animal research conducted in this study comply with Italy's Ministry of Education, University and Research regulations concerning activities and environmental protection in Antarctica and with the Protocol on Environmental Protection to the 137 Antarctic Treaty, Annex II, Art. 3. All the activities on animals performed during the Italian Antarctic Expedition were under the control of a PNRA Ethics Referent, which acts on behalf of the Italian Ministry of Foreign Affairs. In particular, the required data for the project identification code PNRA16_00099 are as follows: Name of the ethics committee or institutional review board: Italian Ministry of Foreign Affairs. Name of PNRA Ethics Referent: Carla Ubaldi, ENEA Antarctica, Technical Unit (UTA).

**Data Availability Statement:** All raw reads and references used for this work are available in public databases and linked with previously published studies, which are referenced in the main text. The results of our analyses are provided as Supplementary Material associated with this manuscript.

**Acknowledgments:** The authors would like to thank Fiorella Florian and Fabrizia Gionechetti for their precious assistance in the wet-lab phase of the experiment. This work was funded by The National Research Program in Antarctica, Grant Nos. PNRA16_00099 and PNRA16_00234 PNRA18_00077.

**Conflicts of Interest:** The authors declare no conflict of interest. The funders had no role in the design of the study; in the collection, analyses, or interpretation of data; in the writing of the manuscript, or in the decision to publish the results.

## Abbreviations

The following abbreviations are used in this manuscript:

| | |
|---|---|
| MDPI | Multidisciplinary Digital Publishing Institute |
| DOAJ | Directory of open access journals |
| TLA | Three letter acronym |
| LD | Linear dichroism |

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
