# Peer review of "Comparative Transcriptomic Analysis Reveals Adaptive Traits in Antarctic Scallop Adamussium colbecki"

_fishes, doi:10.3390/fishes8060276_

Round 1

Reviewer 1 Report

This manuscript conducted comparative transcriptom analysis regarding an Antarctic scallop Adamussium colbecki with four scallop species, Argopecten irradians, Mizuhopecten yessoensis, Azumapecten farreri and Pecten maximus, and tried to explore the possible mechanism of the Antarctic scallop adapting to the freezing environment. Generally, I think the study is interesting and significant since the transcriptom study regarding the Antarctic invertebrates is lacking. However, I suspected the conclusions obtained from current experiment and analysis. Here I listed the weak points and suggestions.

1. Why did the authors use the reads generated with a 3’ tag library preparation protocol? Since the four scallop species used for comparison using the reads generated with a standard poly(A) selection library preparation protocol, I strongly suggested the authors use the same protocol to construct library.

2. Another point I should mention that, in order to get reliable results, it is better to choose relative species or phylogenetic closely related species for cross-species transcriptom comparisons. How the relationships between the four scallop species with the Antarctic scallop A. colbecki? I even suggest the authors choose one or two species for the comparison.

3. There exists quite a lot adaptation studies in Antarctic fish, I hope the author compare their study with them. 

The English writing is sound. 

Author Response

This manuscript conducted comparative transcriptom analysis regarding an Antarctic scallop Adamussium colbecki with four scallop species, Argopecten irradiansMizuhopecten yessoensisAzumapecten farreri and Pecten maximus, and tried to explore the possible mechanism of the Antarctic scallop adapting to the freezing environment. Generally, I think the study is interesting and significant since the transcriptom study regarding the Antarctic invertebrates is lacking. However, I suspected the conclusions obtained from current experiment and analysis. Here I listed the weak points and suggestions.

  1. Why did the authors use the reads generated with a 3’ tag library preparation protocol? Since the fourscallop species used for comparison using the reads generated with a standard poly(A) selection library preparation protocol, I strongly suggested the authors use the same protocol to construct library.

Thank you for this comment. We were fully aware of the potential drawbacks deriving from the different library preparation protocols that were used in the target species and in the other scallops. Therefore, the gene expression quantification strategy used in the present work was the result of multiple preliminary tests, which altogether aimed at minimizing any possible source of bias. First, we need to specify that the 3’-tag libraries from A. colbecki used in this work derive from a different study, aimed at answering a different biological question, and the optimization of sequencing costs, due to the lower sequencing depth required, was the main reason for this choice. This was the subject of another manuscript, which has been accepted in the MDPI’s Stresses journal and is now uploaded as a supplementary file for reviewers’ assessment.

As the reviewer correctly pointed out, the two methods (standard vs 3’-tag library preparation protocols), are expected to lead to drastically different read distributions, making a direct comparison between gene expression level quite difficult. Namely, standard RNA-seq libraries generate reads that are nearly-uniformly distributed along the entire length of each transcript (with a slight bias towards the 3’end in the case of polyA selected libraries), whereas 3’-tag libraries generate reads that mostly map to the few hundred bases close to the 3’-end, i.e. the polyA site. Consequently, read counts obtained from standard libraries require normalization by transcript length, whereas those obtained from 3’-tag libraries do not. Such differences were clearly visible in our study as well, as reported by the mapping distribution shown in Figure 1, which highlights the higher peak obtained towards the 3’end in A. colbecki, compared with the four other scallop species. The graph however also evidences slight differences between these four scallop species with standard RNA-seq libraries available, most likely due to differences in input RNA quality, read length, sequencing platform and library preparation kits.

Our preliminary tests allowed us to identify the final 400 nucleotides of each transcript as the target region linked with the lowest amount of bias in terms of gene expression quantification. Indeed, despite the presence of the aforementioned inter-species differences, the average scaled relative mapping coverage observed in this region was  similar in all species, i.e. 0.67 for A. colbecki, 0.69 for M. yessoensis, 0.72 for P. maximus, 0.75 for A. farreri and 0.79 for A. irradians. This indicates that read mapping distribution did not significantly impact the quantification of gene expression levels, when this window of analysis was used. On the other hand, we calculated higher inter-species differences in mapping rates with alternative mapping strategies. Namely, the expression levels of A. colbecki transcripts would have been systematically under-estimated by using the full length of the transcripts, over-estimated by using a shorter portion of the 3’ ends. Therefore, we believe that this strategy allowed to obtain a reliable comparison of gene expression levels among species.

In any case, since we agree with the reviewer about the importance of clarifying the potential technical biases linked with this study, we moved the paragraph named “limitations” upfront, at the beginning of the discussion section, explicitly stating the presence of these potential issues (rows 260~268). Moreover, the section “Quantification of gene expression” now addresses the mitigation of the issues introduced by using data generated from different protocols (rows 142~149).

  1. Another point I should mention that, in order to get reliable results, it is better to choose relative species or phylogenetic closely related species for cross-species transcriptom comparisons. How the relationships between the four scallop species with the Antarctic scallop A. colbecki? I even suggest the authors choose one or two species for the comparison.

Thank you for this comment. We have updated the text to include more detailed information concerning the phylogenetic relationships between A. colbecki and the four other scallop species (rows 93~99). The four species selected for this study are basically phylogenetically equally distant from A. colbecki, as revealed by a previous phylogenetic analysis performed by our group (see Moro et al. 2019, Marine Genomics). Indeed, all of these belong to the same family (i.e. Pectinidae Rafinesque, 1815), but while A. colbecki belongs to the subfamily Palliolinae Korobkov, 1960, P. maximus and A. irradians are representatives of the subfamily Pectininae, A. farreri and M. yessoensis belong to the subfamily Pedinae Bronn, 1862.

We agree with the reviewer that the most reliable strategy would have been to select non-Antarctic species as phylogenetically close as possible to A. colbecki, i.e. other non-Antarctic members of the Palliolinae subfamily, but this was not possible due to the lack of appropriate –omic resources.

  1. There exists quite a lot adaptation studies in Antarctic fish, I hope the author compare their study with them. 

We are very well aware of this, since we have contributed ourselves to these investigations in Antarctic fish (see Ansaloni et al. 2021, IJMS). We improved the discussion by adding additional references to relevant studies in the field and by highlighting the most significant similarities and differences observed between the adaptation strategies in the Antarctic scallop and notothenioid fishes (rows 286~293, 381~388, 434~436, 488~491).

Reviewer 2 Report

Greco et al. have presented their work entitled Comparative Transcriptomic Analysis Reveals Adaptive Traits In The Antarctic Scallop Adamussium colbecki. I This manuscript is interesting and has provided some information about the adaptation in the Antarctic. In general, this manuscript is well-organized and written. While some issues in this manuscript are required to address before it is accepted for publishing in this journal.

Major comments:

One of my major concerns in this work is the data utilized for the comparative transcriptomic analysis. It seems the data of A. colbecki is quite different from other species examined in the present study as shown in Fig. 1, Fig. 2, and Table S3. It appears that the sequence depth (3,965,333- 9,765,983) of A. colbecki is significantly different from that of the other species examined in the present study (9,377,064- 183,202,268). This issue shall be addressed and discussed.
Mirror comments:

Line 74: No biological information about these species was reported, including body size, sex, etc.

Line 151: What does FDR stand for? More detailed information on this term shall be provided.

Line 163: More detailed information on hypergeometric test shall be provided.

Line 213: figure 4 shall be revised as Figure 4

The quality of English is generally acceptable.

Author Response

Greco et al. have presented their work entitled Comparative Transcriptomic Analysis Reveals Adaptive Traits In The Antarctic Scallop Adamussium colbecki. I This manuscript is interesting and has provided some information about the adaptation in the Antarctic. In general, this manuscript is well-organized and written. While some issues in this manuscript are required to address before it is accepted for publishing in this journal.

Major comments:

One of my major concerns in this work is the data utilized for the comparative transcriptomic analysis. It seems the data of A. colbecki is quite different from other species examined in the present study as shown in Fig. 1, Fig. 2, and Table S3. It appears that the sequence depth (3,965,333- 9,765,983) of A. colbecki is significantly different from that of the other species examined in the present study (9,377,064- 183,202,268). This issue shall be addressed and discussed.

Thank you for this comment, which concerns a similar point to the one raised by reviewer #1. This discrepancy was due to the different library preparation strategy we used compared to the other studies. In detail, we prepared 3’-tag libraries (using the kit provided by Lexogen), which, compared with standard RNA-seq libraries, generates fragments enriched in the 3’end of each transcript, close to the poly-A site. The main advantage of this protocol lies in the significantly reduced costs per sample, due to the lower required sequencing depth, i.e. approximately 10% than what would be needed to achieve an equally reliable estimate of gene expression levels using a standard library. This is due to the different distribution of reads in standard libraries, which would be expected to cover the entire length of a transcript.

The choice for the use of this strategy was made to address the primary biological question our studied on A. colbecki was focused on, i.e. its ability to respond to thermal stress. This was the subject of a companion paper, that has already been accepted for publication on the journal Stresses and is planned to be published back-to-back with this one, pending its acceptance. We have uploaded as a supplementary material the full text of this accepted manuscript for the reviewer’s assessment.

Hence, while the lower sequencing depth used in this study was not detrimental for assessing gene expression levels, we recognize that this approach may lead to technical biases, which we tried to appropriately address, for example by calculating gene expression levels based on the reads mapped on the last 400 nt of each transcript only, as now discussed in detail in the revised version of this manuscript (in materials and methods at rows 142~149 and in discussion at rows 260~268) (see also the response to reviewer #1).

Mirror comments:

Line 74: No biological information about these species was reported, including body size, sex, etc.

This information is provided in detail in the companion manuscript, which is now appropriately referenced in the materials and methods section. As far as the four other scallop species are concerned, detailed information concerning sampled animals can be found in the manuscripts referenced in the main text (rows 76, 88~89).

Line 151: What does FDR stand for? More detailed information on this term shall be provided.

We provide this term in full on its first mention, i.e. False Discovery Rate (rows 166~167).

Line 163: More detailed information on hypergeometric test shall be provided.

We added a reference to a manuscript detailing the methodology (rows 180~181).

Line 213: figure 4 shall be revised as Figure 4

Thank you, this was amended.

Round 2

Reviewer 2 Report

I appreciate that the authors have made a lot of efforts to address my previous comments on the manuscript. Yet, I am not fully convinced that it is appropriate to perfom comparative studies using transcriptomic data of different species based on different library perpration. Please provide the reference to support it is appropriate to do so. 

Besides, some issues are still present in the latest manuscript. 

Line 48: usually "70 75" mm long ?

Line 48: are you capable "of of" ?

Line 56: "plicae" ?

Liune 66: fish [20]","little ?

Line 67: amount of "-"omic resources

Line 97-99: The taxonomic description is not correct. M.yessoensis and A. farreri are "NOT" members of Pectininae. P. maximus and A. irradians are "NOT" members of Pedinae. <=Inconsist with the reference you cited.

Line 106: 1 × 10−10 and 80%" "respectively.

Line 260-279: More references are required to justifiy your statement.

Author Response

I appreciate that the authors have made a lot of efforts to address my previous comments on the manuscript. Yet, I am not fully convinced that it is appropriate to perfom comparative studies using transcriptomic data of different species based on different library perpration. Please provide the reference to support it is appropriate to do so. 

We thank the reviewer for his assessment and tried to further address his/her concerns in this second revised version of the manuscript. The potential issues pointed out by the reviewer are similar to those we initially identified while planning this experiment and analyzing the data, which led us to carry out the different preliminary tests explained in the previous version of this revision. As correctly identified by the reviewer, the issues linked with this comparative inter-species analysis can lead to systematic biases, hampering a correct interpretation of gene expression data. These issues lie on two distinct levels:

1) the possibility to compare gene expression profiles between different species (e.g. because the structure of the transcriptome is different, due to the presence of a different number of gene copies, etc.)

2) the possibility to compare gene expression profiles obtained using different library preparation protocols (e.g. because read mapping distribution impacts the quantification of gene expression levels)

We will briefly address below the scientific consensus about the impact of these two factors, reporting in more detail in the text how we ensured to mitigate their effects in our study.

1) Numerous studies support the feasibility of inter-species transcriptomic comparisons. Nevertheless, these approaches require a careful design and need to acknowledge the existence of limitations, since they need to be focused on a set of genes shared by all target species to allow full comparability among gene expression profiles. For example, we have demonstrated that the comparison of gene expression profiles between Antarctic and non-Antarctic fish species led to reproducible results (see Ansaloni et al. 2021, IJMS), as long as such comparisons are limited to transcripts that could be unequivocally identified as orthologs. The idea of basing inter-species gene expression comparisons on orthologous genes is indeed quite old, as it was first suggested by Grigoryev and colleagues back in 2004 (see https://doi.org/10.1186%2Fgb-2004-5-5-r34) and later applied to microarray studies (e.g. Kristiansson et al. 2013, see https://doi.org/10.1186/1471-2105-14-70). The application of the same approach to RNA-seq data is, for  example, supported by the implementation of OrthoMCL in the recently published CoRMAP pipeline, specifically designed for comparative transcriptomic studies (see Sheng et al. 2022, https://doi.org/10.1186/s12859-022-04972-9, now also cited in the revised manuscript). Other recent significant examples of comparative transcriptome studies that used orthologous transcript detection are Vercruysse et al. 2020 (between Arabidopsis and maize, see https://doi.org/10.1111/pbi.13223), or Saxena et al. 2022 (between mouse and jerboas, see https://doi.org/10.1016/j.cub.2021.10.063). In summary, to answer to the first part of the question posed by the reviewer (i.e. “Is the possibility to carry out inter-species transcriptomic comparisons between species supported by literature?”): yes, this is possible, with the limitation of carrying out such comparisons on transcripts identified as orthologous with high confidence. As explained in the main text, we ensured this was the case, by applying a rigorous methodology for the detection of orthology relationships among the different species.

2) A few studies have been specifically carried out to comparatively evaluate the performance between the two library preparation approaches (standard libraries vs 3’-tag libraries). Despite the different mapping profiles they produce, a few studies carried out to comparatively evaluate their performance highlighted a high correlation between gene expression levels (i.e. correlation coefficient was = 0.85 in Ma et al. 2019, see https://doi.org/10.1186/s12864-018-5393-3, and = 0.75 in Xiong et al. 2017, see https://doi.org/10.1038/s41598-017-14892-x). Obviously, the two approaches have advantages and disadvantages, but literature is concordant in attributing a high consistency between the results produced with the two methods. We have no reason to believe that this correlation between the two methodologies would be any different in our study. Hence, although a few outlier genes may be detected as differentially expressed due to technical discrepancies between the two library preparation protocols (as now acknowledged in the text), we do not expect these to significantly influence the general trends which led to the generalized overexpression of a few gene families highlighted by the functional enrichment analysis. Please also note that, in order to further mitigate the impact of the different read mapping distribution linked with the two library preparation protocols, we considered only the final 400 nucleotides of each transcript for calculating gene expression levels, as this region was the one to show the lowest inter-species differences in overall coverage (i.e. the average scaled relative mapping coverage observed in this region was  similar in all species, i.e. 0.67 for A. colbecki, 0.69 for M. yessoensis, 0.72 for P. maximus, 0.75 for A. farreri and 0.79 for A. irradians). Consequently, to answer the second part of the question posed by the reviewer (i.e. “Is the possibility to carry out transcriptomic comparisons between samples sequenced using different library preparation protocols supported  by literature?”: yes, this is possible, as correlation coefficients are generally very high, but caution should be taken since a few outlier genes may display high variations due to technical factors. This is now more explicitly stated in the discussion section.

Despite this, the main concern expressed by the reviewer is justified: as a matter of fact, there are no studies that, to the best of our knowledge, have tried to address the combined effects of both factors and therefore we cannot provide any specific reference to studies that have previously used the very same approach we propose. In light of this, we need to be very straightforward about the limitations of this study and we believe that the long section included in the discussion serves this purpose. As an additional data to support the comparability between the gene expression profiles obtained in A. colbecki and the other scallop species we provide the correlation matrices obtained through all pairwise comparisons in the three tested tissues. Such tables report summarize the expression values of all orthologous genes (excluding those lacking expression, and thereby undetectable) in A. colbecki. Here, correlations coefficients among non-Antarctic species (with libraries constructed using the same method) were 0.85 ± 0.13, whereas those observed between A. colbecki and non-Antarctic scallops (with different library preparation protocols) were slightly lower, i.e. 0.70 ± 0.08. Such correlation coefficient was just slightly lower than those reported in literature by Xiong et al. 2017 (i.e. 0.75) between replicates obtained starting from the same RNA using the two library preparation approaches. Since we are not very distant from these expectations, we interpret the marginal difference to be the likely result of real biological differences linked with the adaptation to the Antarctic environment or to the occasional presence of outliers showing a lower apparent expression value in A. colbecki (please note that down-regulated genes were disregarded, since we were well aware of this potential issue). Such correlation tables are provided as supplementary materials for an assessment by the reviewer.

Besides, some issues are still present in the latest manuscript. 

Line 48: usually "70 75" mm long ?

Thank you for pointing out this. We intended to indicate the most usual range of length, so this was modified to “70 to 75 mm long”.

Line 48: are you capable "of of" ?

Thank you, this was corrected.

Line 56: "plicae" ?

“Plicae” is a technical term used to design the rib-like structures that characterize the surface of pectinid shells. To avoid confusion, we replaced this term with “radial ribs”. The term was also incorrectly written in italics.

Liune 66: fish [20]","little ?

A space was missing. This was added.

Line 67: amount of "-"omic resources

With this term, we meant to indicate different types of –omic resources, such as transcriptomics and genomics. We recognize that the use of this term may lead to some confusion in the reader, so we chose to replace it with “molecular resources”.

Line 97-99: The taxonomic description is not correct. M.yessoensis and A. farreri are "NOT" members of Pectininae. P. maximus and A. irradians are "NOT" members of Pedinae. <=Inconsist with the reference you cited.

Thank you for pointing this out! We mistakenly exchanged the association of the four species to their respective subfamilies.

Line 106: 1 × 10−10 and 80%" "respectively.

We checked the indications for reporting numbers with scientific notation based on the authors’ guidelines for this journal. To simplify the understanding of this sentence, it was modified as follows: “setting the evalue threshold to 1 × 10−10 and the percentage of identity threshold to 80%”.

Line 260-279: More references are required to justifiy your statement.

This section was improved by adding several of the references reported in the first answer to reviewer’s comments.